# Ameliorative Effects of Peptides from the Oyster (*Crassostrea hongkongensis*) Protein Hydrolysates against UVB-Induced Skin Photodamage in Mice

**DOI:** 10.3390/md18060288

**Published:** 2020-05-31

**Authors:** Zhilan Peng, Beibei Chen, Qinsheng Zheng, Guoping Zhu, Wenhong Cao, Xiaoming Qin, Chaohua Zhang

**Affiliations:** 1College of Food Science and Technology, Guangdong Ocean University, Zhanjiang 524088, China; pengzhilan@stu.gdou.edu.cn (Z.P.); chenbeibei4@stu.gdou.edu.cn (B.C.); zqs@stu.gdou.edu.cn (Q.Z.); zjougp@gdou.edu.cn (G.Z.); cwenhong@gdou.edu.cn (W.C.); qinxm@gdou.edu.cn (X.Q.); 2Guangdong Provincial Key Laboratory of Aquatic Products Processing and Safety, Zhanjiang 524088, China; 3Guangdong Province Engineering Laboratory for Marine Biological Products, Zhanjiang 524088, China; 4National Research and Development Branch Center for Shellfish Processing (Zhanjiang), Zhanjiang 524088, China; 5Key Laboratory of Advanced Processing of Aquatic Product of Guangdong Higher Education Institution, Zhanjiang 524088, China

**Keywords:** polypeptides, oyster, photoaging, antioxidative, anti-inflammatory

## Abstract

Chronic exposure to ultraviolet B (UVB) irradiation is a major cause for skin photoaging. UVB induces damage to skin mainly by oxidative stress, inflammation, and collagen degradation. This paper investigated the photo-protective effects of peptides from oyster (*Crassostrea hongkongensis*) protein hydrolysates (OPs) by topical application on the skin of UVB-irradiated mice. Results from mass spectrometry showed that OPs consisted of peptides with a molecular weight range of 302.17–2936.43 Da. In vivo study demonstrated that topical application of OPs on the skin significantly alleviated moisture loss, epidermal hyperplasia, as well as degradation of collagen and elastin fibers caused by chronic UVB irradiation. In this study, OPs treatment promoted antioxidant enzymes (SOD and GPH-Px) activities, while decreased malondialdehyde (MDA) level in the skin. In addition, OPs treatment significantly decreased inflammatory cytokines (IL-1β, IL-6, TNF-α) content, and inhibited inflammation related (iNOS, COX-2) protein expression in the skin. Via inhibiting metalloproteinase 1(MMP1) expression, OPs treatment markedly decreased the degradation of collagen and elastin fibers as well as recovered the altered arrangement of extracellular matrix network in the dermis of skin. Our study demonstrated for the first time that OPs protected against UVB induced skin photodamage by virtue of its antioxidative and anti-inflammatory properties, as well as regulating the abnormal expression of MMP-1. The possible molecular mechanism underlying OPs anti-photoaging is possibly related to downregulating of the MAPK/NF-κB signaling pathway, while promoting TGF-β production in the skin.

## 1. Introduction

Solar ultraviolet (UV) radiation is one of the most dominant threats responsible to the skin damage. According to its wavelength, UV light can be subdivided into UVC (200–280 nm), UVB (280–315 nm), and UVA (315–400 nm) regions. Long-term exposure to ultraviolet (UV) radiation, especially directly exposed to UVA and UVB, is known to damage the structural integrity and physiological function of the skin [1]. This damage is referred collectively as skin photoaging, which is characterized by the development of coarse solar scars, brown spots, drying, irregular pigmentation, reduced elasticity, formation of wrinkles, and a leathery appearance [2,3,4]. Due to its high energy and short wavelength characters, UVB could pass through the epidermis and penetrate the upper part of the dermis, and it is the predominant cause for skin photoaging [5,6].

UVB irradiation not only directly damages biological macromolecules, such as deoxyribonucleic acid (DNA), lipids, and proteins, but also decreases the activities of antioxidant enzymes, such as superoxide dismutase (SOD) and glutathione peroxidase (GSH-Px), increasing the relative oxygen species (ROS) production in the skin [7]. By activating mitogen-activated protein kinase (MAPK) signaling pathways, UVB induced ROS production on the one hand promotes the activation of the transcription factors NF-κB, which further stimulates the production of inflammatory cytokines, such as interleukin (IL)-1β (IL-1β), IL-6, and tumor necrosis factor-α (TNF-α) [8,9]. On the other hand, UVB radiation or UVB induced-ROS production stimulate the expression of cyclooxygenase-2 (COX-2), which upregulates the production of inflammatory mediators such as inducible nitric oxide synthase (iNOS) [10,11]. These inflammatory mediators and cytokines will further accelerate the accumulation of reactive oxygen species (ROS) that damage skin tissues and cells, and induce collagen degradation via enhancing the expression of various matrix metalloproteinases (MPPs), such as MMP-1, MMP-3, and MMP-9, hastening the skin aging process [11,12]. In particular, MMP-1 overexpression was reported to initiate the degradation of transforming growth factor beta (TGF-β), collagen types I, and elastin in the skin [13,14]. Thus, anti-oxidative stress, anti-inflammation, as well as inhibiting metalloproteinases, are main strategies for the prevention and treatment of UVB-induced skin photoaging.

Hong Kong oyster (*Crassostrea hongkongensis*) is an economically important invertebrate found in mudflats and widely cultivated in southern China [15]. Around 400 years ago, it was recorded in ‘Ben Cao Gang Mu’ that oyster helps in whitening and smoothing skin. Meanwhile, numerous studies showed that oyster and oyster-derived peptides possess multiple activities such as prevention of hyperglycemia [16], anti-apoptotic [17], antioxidant [18], immunomodulatory [19], and anti-inflammatory effects [20]. In previous studies, Jae Hyeong Han et al. reported that the oral administration of Pacific oyster (*Crassostrea gigas*) hydrolysate exerted anti-melanogenic effects in UVB-irradiated C57BL/6J mice, as well as protected against wrinkle formation in UVB-irradiated hairless mice [21,22]. A pentapeptide derived from oyster hydrolysate was also showed effect on reducing wrinkles [23]. However, people paid little attention to the benefits of the topical application of Hong Kong oyster-derived peptides on the UVB-induced skin photoaging damage.

Therefore, in this study, peptides (OPs) from the oyster (*Crassostrea hongkongensis*) protein hydrolysates were obtained, and their characteristics determined. Then, the anti-photoaging effects of OPs were evaluated by macroscopic observation and three histological staining methods on the skin of the UVB-irradiated Kunming mice. The possible mechanism underlying its photo-protective effects were studied by measuring the activities of antioxidant enzyme (SOD, GSH-Px), malondialdehyde (MDA) production, and hydroxyproline (HYP) content with commercial kits, evaluating the expression of inflammatory cytokine factors (IL-1β, IL-6, TNF-α) with enzyme-linked immunosorbent assay (ELISA) kits, and investigating the expressions of MMP-1 and COX-2 in the skin by immunohistochemical staining. Eventually, the MAPK/ NF-κB signaling pathway, inflammation related (iNOS, COX-2) protein expression, as well as the most effective collagenase MMP-1 activity involved in skin photoaging process were further investigated by Western blotting. The results from this study may provide a strong foundation for the application of OPs as therapeutic or cosmetic products.

## 2. Results

### 2.1. Molecular Weight Distribution and Main Peptide Sequences of OPs

Figure 1 provided the total ion mass spectrum of the purified OPs, as determined by primary mass spectrometry, along with the overall molecular weight distribution. As shown in Figure 1, the OPs peaks variously appeared from 300 to 700 m/z, which identified a total of 687 peptides in the molecular weight range of 302.17–2936.43 Da. Analysis results from comparing the UniProt database (www.uniprot.org/) with Max quant (1.6.2.10) demonstrated that OPs mostly consist of peptides with molecular weights below 2000 Da, which accounted for 95.6% of total peptides. Peptide fingerprinting of 24 characteristic peptides in OPs was analyzed using a liquid chromatography-mass spectrometry (LC-MS/MS). Further analysis results demonstrated that OPs mainly contained 24 peptide chains 4–17 amino acids in length, as shown in Table 1. It is noteworthy that among 24 characteristic peptides, there are amino acids with higher antioxidant activity (such as Cys, Met, Trp), and the disulfide bond in the amino acid structure exhibits strong reducibility.

### 2.2. Amino Acid Composition of OPs

According to the data from the automatic analyzer, the amino acid composition and content (g/100 g total amino acids) of OPs are shown in Table 2. Among the 18 amino acids found in OPs were glutamic acid (3.59 g/100 g), aspartic acid (2.18 g/100 g), arginine (1.87 g/100 g), glycine (2.07 g/100 g), and alanine (1.70 g/100 g). In this study, essential amino acids accounted for 31.25% of the total amino acids, hydrophobic amino acids accounted for 43.81% of the total amino acids, and hydrophilic amino acids accounted for 31.48% of the total amino acids.

### 2.3. Morphology and Histological Observation of OPs Photo-Protective Effects on UVB-Induced Skin Photodamage

During the 10-week period of experiment, there were no significant differences in the average body weight between groups (as shown in Appendix A).

#### 2.3.1. Macroscopic Observation Results

Macroscopic effects of UVB irradiation on mouse dorsal skin are shown in Figure 2. As expected, the dorsal skin of mice in the CT group appeared healthy throughout the whole study without any erythema or winkling. Conversely, after 10 weeks of UVB irradiation, the dorsal skin of mice in the UVB group developed distinct photoaging features: dark-brown color, deep-wrinkled, sunburns, and leathery skin with some flesh-colored lesions (Figure 2A). Although the appearance of skin was improved a little by low dose of OPs application, it was still brown, and with clear signs of UVB irradiation. However, adverse appearances of dorsal skin caused by UVB irradiation were significantly improved in mice whose skin had been treated with middle and high dose of OPs, the effects of which were even better than the positive agent vitamin C treatment. The skins in the high dose application of OPs group were noticeably smoother, less wrinkly, and showed no signs of melasma and sunburns, which were almost close to the normal group. Further evaluation of moisture contents indicated that the moisture content of dorsal skin (Figure 2C) was significantly decreased after 10 weeks of UVB exposure (*p* < 0.01). Topical application of OPs markedly ameliorated the moisture loss of photodamaged skin in a dose-dependent manner (*p* < 0.05 for the middle dose group, and *p* < 0.01 for the high dose group).

#### 2.3.2. HE Staining Results

HE staining results displayed that the dorsal skin of control group appeared healthy, with no obvious abnormal structures (Figure 2B). Compared with the control group, the skin of mice exposed to UVB irradiation developed a thickened epidermis, irregularly arranged collagen fibrils, broken nuclei, and inflammatory cell infiltration of the dermis. However, the changes of UVB-irradiated skin could be attenuated by OPs topical application in a dose-dependent manner, as well as VC treatment. As we can see from Figure 2B, compared with UVB group, the dorsal skin of mice in the UVB+OPs-H group exerted thin cortex, regularly arranged collagen fibrils, as well as plump subcutaneous hair follicles and sebaceous glands, which was almost close to the normal group. Skinfold, one of the important features of photoaging, was also showed in Figure 2B, and quantified in Figure 2D with measuring the thickness of epidermis of dorsal skin. In this study, UVB irradiation significantly (*p* < 0.01) increased the thickness of epidermis from 18.5 μm in the control group to 54.8 μm in the UVB group. However, UVB-induced epidermal hyperplasia was significantly attenuated by topical application of OPs on the skin with either middle or high dose, as well as VC administration (*p* < 0.01, respectively).

#### 2.3.3. Masson Trichrome Staining Results

The effect of OPs on the UVB-induced damage of dermal collagen fibers was assessed by Masson trichrome staining (Figure 3). In the control group, collagen fibers in the dermis of dorsal skin were stained abundantly with blue dye. However, after 10 weeks of UVB exposure, collagen fibers stained with blue dye were dramatically decreased, meanwhile collagen bundles in the dermis of the skin were distorted, broken, and even abnormally accumulated. The reduction of collagen fibers caused by UVB exposure was prominently ameliorated by OPs topical application on the dorsal skin with either middle or high dose (*p* < 0.05, *p* < 0.01, respectively), as well as VC treatment (*p* < 0.01). In particular, the structure and density of dermal collagen fibers in the UVB +OPs-H group was almost closed to that in the normal group.

#### 2.3.4. Victoria Blue Staining Results

The effect of OPs on UVB-induced damage of elastin fibers in the dermis of dorsal skin was evaluated by using Victoria Blue staining. As shown in Figure 4, the elastin fibers in the dermis appeared reticular structure, with no abnormal stacked debris. After UVB exposure, elastin fibers stained with pale blue dye were apparently decreased, elastic fibers were arranged disorderly, and the elastin network structure were cracked into pieces. However, the reduction and disordered arrangement of elastin fibers induced by UVB irradiation were ameliorated by high dose topical application of OPs (especially with the dose of 0.8 g/kg∙bw) on the skin. The contents and arrangements of elastin fiber in the UVB + OPs-H group were almost close to that of the normal group.

### 2.4. OPs Protect Antioxidant Enzyme (SOD, GSH-Px) Activity, Inhibit MDA Production, and Eventually Promote Collagen Contents in the Dorsal Skin

As shown in Table 3, compared with the control group, the activities of antioxidant enzyme were significantly decreased after 10 weeks of UVB continuous radiation. For instance, the antioxidant activities in the UVB group decreased by 50.0% for SOD, and by 60.4% for GSH-Px (*p* < 0.05 and *p* < 0.01, respectively). However, decreased activities of SOD and GSH-Px could be improved by VC administration, as well as topical application of OPs with either middle or high doses. Notably, UVB induced decrease in SOD and GSH-Px activities were completely reversed by topical application of OPs at high dose (*p* < 0.01).

Increased MDA level is an important index of oxidative stress. After 10 weeks of UVB irradiation, MDA concentration in the dorsal skin of mice was significantly increased (*p* < 0.01). However, this increase in MDA content after UVB exposure could be attenuated by topical application of OPs at middle or high dose, as well as VC treatment (*p* < 0.01, respectively). Moreover, the effect of OPs anti-lipid oxidation was much better than that of VC, as the MDA level in the UVB+OPs-L group was even lower than that in UVB+VC group.

In line with results from morphology and histological observation, compared with the CT group, collagen content in the dorsal skin of mice was significantly decreased in UVB group (*p* < 0.01). However, UVB-induced collagen level decrease could be significantly improved by OPs application at either middle or high doses, but not VC treatment (*p* < 0.05, respectively).

### 2.5. OPs Decrease Pro-Inflammatory Factor (IL-1β, IL-6, TNF-α) Contents in the Skin

As shown in Figure 5, UVB exposure significantly elevated the concentrations of IL-1β, IL-6, and TNF-α in the skin of mice (Figure 5). In this study, VC treatment showed limited anti-inflammatory effect, as only TNF-α expression was inhibited by VC topical application (*p* < 0.05; Figure 5C). However, middle dose topical application of OPs decreased IL-6 and TNF-α concentrations in the UVB+SPMs-M group (*p* < 0.01 respectively; Figure 5B,C), and high dose treatment with OPs significantly downregulated the IL-1β, IL-6, and TNF-α levels in the UVB+SPMs-H group (*p* < 0.01 respectively; Figure 5A–C).

### 2.6. OPs Downregulate MMP-1 and COX-2 Expressions in the Skin, Detected by Immunohistochemical (IHC) Staining

As we can see from Figure 6, compared with control group, the expressions of COX-2 in the epidermis and MMP-1 in the dermis were significantly increased after UVB exposure. Elevated COX-2 level in the epidermis could by ameliorated by OPs topical application on the skin at high dose, as well as VC administration (*p* < 0.01 and *p* < 0.05, respectively) (Figure 6A,C). In a similar pattern, increased MMP-1 levels in the dermis could be downregulated by OPs treatment in a dose-dependent manner, as well as VC treatment. Particularly, MMP-1 expression in the UVB + OPs-high group was almost recovered to that in the control group (Figure 6B,D).

### 2.7. OPs Regulate MAPK Signaling and NF-κB in UVB-Irradiated Mice Skin

Inflammation-mediated skin aging is a major factor in UVB-induced skin photoaging, which could be initiated by UVB direct exposure or the huge production of ROS that arises from UVB-induced oxidative stress via activating MAPK signaling proteins in the skin. Activated MAPK signaling could further stimulate nuclear factor-κB (NF-κB) to promote the productions of inflammatory cytokines in the skin, which contributes to the skin photoaging.

As shown in Figure 7, MAPK signaling proteins of extracellular signal-regulated kinases (ERKs), c-Jun N-terminal kinases (JNKs), and p38 were activated after 10 weeks of UVB exposure (*p* < 0.01, respectively). The upregulated p-ERK and p-JNK phosphorylation were significantly inhibited by topical application of OPs at high dose, and VC treatment (Figure 7B,D). Similarly, the activation of p38 MAPK could be downregulated by topical application of OPs on the skin in a dose-dependent manner, as well as VC administration (*p* < 0.01, respectively; Figure 7C).

Further analysis study revealed that NF-κB (p65) expression was activated by MAPK signaling protein phosphorylation (*p* < 0.01) in the skin. However, NF-κB (p65) activation could be dramatically downregulated by the topical application of OPs on the skin at either middle or high dose, as well as VC administration (*p* < 0.01, respectively).

### 2.8. OPs Inhibit Inflammatory-Related Proteins (COX-2 and iNOS) Expression in Dorsal Skin Samples

To further explore the possible mechanisms of OPs anti-photoaging, protein expressions of pro-inflammatory enzymes (COX-2 and iNOS) in dorsal skin were further studied. As shown in Figure 8, 10 weeks of UVB irradiation induced a significant increase in the protein expressions of COX-2 (*p* < 0.01) and iNOS (*p* < 0.01). However, this increase in COX-2 and iNOS protein expression could be attenuated by middle- or high-dose topical application of OPs on the skin, as well as VC treatment (Figure 8A).

### 2.9. OPs Control the Level of MMP-1, TGF-β in UVB-Exposed Mice

According to the Western blotting result shown in Figure 8B, compared with control group, the expression of MMP-1 in the dorsal skin of mice exposed to UVB was significantly increased. However, UVB-induced MMP-1 increase could be dramatically inhibited by the topical application of OPs at middle (*p* < 0.01) or high (*p* < 0.01) dose, as well as VC treatment (*p* < 0.01). In contrast, the expression of TGF-β in the dorsal skin was significantly decreased after UVB irradiation, while this decrease in TGF-β expression could be significantly upregulated by topical application of OPs in a dose-dependent pattern (*p* < 0.05, *p* < 0.01, *p* < 0.01, respectively).

## 3. Discussion

Previous studies had found that oral administration of pacific oyster (*Crassostrea gigas*) hydrolysates protected against UV-induced skin damage [21,22]. However, this method of applying oyster may limit the absorption of macromolecular nutrients, such as peptides [24]. At the same time, increasing evidences indicated that topical application of peptides on skin would hydrate the stratum corneum, increase permeability, and enhance drug absorption, eventually shortening the reaction time [24,25]. Thus, it is important to evaluate the ability and efficiency of topical application of peptides from the protein hydrolysates of oyster (*Crassostrea hongkongensis*) in a UVB-induced photodamaged mouse model. In the present study, we demonstrated that topical treatment of OPs significantly alleviated moisture loss, epidermal hyperplasia, as well as degradation of collagen and elastin fibers on the dorsal skin caused by chronic UVB irradiation. The strength and resilience of skin is strongly influenced by the structural organization and functions of collagen and elastin fibers [26,27]. The degeneration and fragmentation of collagen and elastin fibers could lead to skin wrinkles and sagging, which is an important symptom of skin photoaging [28,29]. In this study, topical application of OPs on skin inhibited UVB irradiation induced-abnormal arrangement of collagen and elastin fibers in the dermis. The results from the present study suggested that OPs could prevent UVB-induced collagen and fiber damage.

It is well established that the degradation of collagen and elastin fiber network in the skin was promoted by the production of matrix metalloproteinases (MMPs), a group of zinc-dependent extracellular proteinases [9,30]. In particular, MMP1 (interstitial collagenase-1, the most effective collagenase) could degrade the major structural protein (collagen types I and III), and elastin in the skin, thereby causing the wrinkling and sagging of photo-damaged skin [9,31]. In this study, as shown in both IHC and WB results, the expression of MMP-1 was excessively increased in UVB-irradiated mice skins, thereby the skin displayed an obviously wrinkled appearance macroscopically and exhibited tangled and degraded collagen and elastic fibers histo-pathologically. However, topical OPs application on the skin (especially with the dose of 0.8 g/kg∙bw) significantly suppressed the UVB-induced increase in MMP-1 production, while elevating the levels of collagen and elastin in the dermis of the skin. In addition, compared with the mice exposed to UVB irradiation, the skin of mice with OPs topical application exhibited better distributed collagen and elastic fibers, as well as a greater visually compact and smooth appearance, which is in line with the results observed in the macroscopic evaluation and histological study. In order to investigate the mechanism by which OPs so greatly increases collagen and elastin levels, we measured the protein expression of TGF-β, as the TGF-β/Smad pathway is a major pathway controlling procollagen production [32]. We found that OPs significantly decreased MMP-1 production, while promoting the TGF-β protein expression in UVB-irradiated skin. Thus, OPs mitigate the degradation of collagen and elastic fibers possibly via downregulating expression of MMP-1, while increasing TGF-β production in the skin.

As mentioned above, it is well reported that MMPs can be promoted by both reactive oxygen species and inflammatory cytokines. Evidence demonstrated that UVB irradiation could decrease the activities of antioxidant enzymes, such as SOD and GPH-Px [29], and promote oxidative damage in the skin [7]. In a previous study, peptides from oyster (*Saccostrea cucullata*) protein hydrolysate was reported to possess good antioxidant properties [33], indicating OPs have potential for the prevention of photoaging. Here, we demonstrated that after 10 weeks of UVB exposure, mouse dorsal skin displayed a notable decrease in activities of antioxidant enzymes. However, topical OPs (especially with the dose of 0.8 g/kg∙bw) application on skin significantly elevated activities of these enzymes, indicating that OPs exerted favorable effects against skin oxidative damage induced by UVB irradiation. It is well known that UV-induced ROS production can promote the increase of lipid peroxidation (LPO) by reacting with unsaturated fatty acids in the cell membrane, which eventually results in a decrease of collagen content [34,35,36]. In line with previous studies, exposure of the skin to UVB radiation for 10 weeks caused an obvious increase in MDA levels (a stable end-product of LPO), but a decrease of collagen content in the skin. In this study, topical OPs application (especially with the dose of 0.8 g/kg∙bw) significantly downregulated the increased MDA level induced by UVB exposure. We, therefore, concluded that OPs possess the ability to inhibit MMPs secretion possibly via protecting endogenous antioxidative enzymes, as well inhibiting MDA production in the skin.

Previously, UVB-induced ROS production could activate MAPK signaling to induce inflammation and skin aging by increasing the phosphorylation of p38, JNK, and ERK (p-p38, p-JNK, p-ERK) [32]. MAPK signaling activation may further stimulate nuclear factor-κB (NF-κB) to activate the gene transcription and protein expression of iNOS and COX-2, thereby promoting the production of inflammatory cytokines in the skin [37,38]. In this study, the results from HE staining and Western blotting indicated that inflammatory cells infiltration in the skin of mice exposed to UVB irradiation was accompanied with MAPK/ NF-κB signaling activation, as well as the increase of iNOS and COX-2 expressions in the skin. These over-expressed cytokines negatively alter the function of skin cells, and undesirably impact the metabolism of extracellular matrixes, such as collagen and elastin fibers [39,40]. Previous studies have also reported that inflammatory cytokines, such as TNF-α and IL-1β, are key mediators in the early stage of sunburn response that can induce apoptosis, while IL-6 is a common inflammatory marker associated with chronic diseases [29,41]. As shown in this, and previous studies [29,32], inflammatory cytokines could activate MMP1 to accelerate the degeneration of collagen and elastin fibers, thereby promoting photodamage. Therefore, administration of anti-inflammatory agents may protect skin against photodamage. In this study, we found that topical OPs (especially with a dose of 0.8 g/kg∙bw) application strongly decreased the over-production of inflammatory cytokines (IL-1β, IL-6, TNF-α) and iNOS and COX-2 in the skin. Therefore, the protective effect of OPs in UVB-induced skin damage may be related to its anti-inflammation properties via downregulating the production of inflammatory mediators and cytokines, as well as inhibiting the inflammatory response associated MAPK/NF-κB signaling pathway.

In summary, our study demonstrated for the first time that OPs exhibited prominent photoprotection activity by virtue of its antioxidative and anti-inflammatory properties, as well as regulating the abnormal expression of MMP-1. The possible molecular mechanism underlying OPs anti-photoaging is related to downregulating of MAPK/ NF-κB signaling pathway, while promoting TGF-β production in the skin. A schematic explanation for the UVB-mediated photoaging and the role of OPs has been shown in Figure 9. This study provided an available pre-clinical evidence for the application of OPs as a therapeutic and cosmetic product against skin photoaging.

## 4. Materials and Methods

### 4.1. Preparation of Peptides from Oyster Protein Enzymatic Hydrolysates (OPs)

Hong Kong oyster (*Crassostrea hongkongensis*) was obtained from the adjacent sea of Zhanjiang City in China. Firstly, after removing all visible impurities and debris, fresh oyster meat (10 kg for this study) was homogenized with three volumes of deionized water. Then oyster protein was extracted by pH-shifting method (alkali-soluble and acid precipitation conditions of the oyster protein isolate were pH = 12 and pH = 4.8, respectively), and a total of 658.8 g of oyster protein was collected in this step. Subsequently, after reconstitution in three volumes of distilled water, the freeze-dried oyster protein was hydrolyzed for 3 h at 50 °C (pH = 7.0) by the addition of 2% neutral protease (Pangbo Biotech, Nanning, China, 2 × 10^5^ U/g). Followed inactivation steps, enzymatic hydrolysate was centrifuged at 12,000 rpm for 20 min. Finally, supernatants were gradually fractionated by 8 kDa, 5 kDa, and 3 kDa ultrafiltration membranes (Amicon^®^ Ultra-15 centrifugal filter units, Billerica, MA, USA) to separate peptides we used in this study (< 3 kDa hydrolysate fractions). In total, 55.2 g of freeze-dried OPs were harvested for further mass spectrometry and in vivo study.

### 4.2. Determination of Molecular Weight Distribution

The molecular weight distribution of OPs was identified by our previous methods with slight modification [42]. Firstly, OPs were subjected to reductive alkylation treatment, and then desalted by a self-priming desalting column before determination. Subsequently, alkylated and desalted OPs sample was isolated by using ultra higher performance liquid chromatography (UHPLC) (ultimate 3000; Thermo Fisher Scientific Inc., Waltham, MA, USA) packed with an Acclaim Pep Map RPLC C18 (Φ75 μm id × 150 mm) column (Dionex™; Thermo Scientific). The mobile phase was composed of 0.1% formic acid (FA)/2% ACN (solvent A) and 0.1% FA/80% CAN (solvent B). The gradient condition was 6% B to 40% B in 75 min (and held for 3 min at 95% B), the flow rate was 300nL/min, and the wavelength of the ultraviolet detector was 220 nm. Eventually, the molecular mass distribution of OPs was identified by electrospray ionization mass spectrometry and tandem mass spectrometry (ESI-MS/MS) in a positive ion mode. The formulation of LTQ VELOS ESI cation calibration solution was following previous study [42]: caffeine (2 µg/mL), MRFA (1 µg/mL), Ultramark 1621 (0.001%), and n-butylamine (0.0005%) in acetonitrile (50%), aqueous methanol (25%), and acetic acid (1%).

### 4.3. Major Peptide Sequence Analysis of OPs

According to previous methods by our group and other researchers [42,43], peptide sequences were identified by electrospray ionization mass spectrometry and tandem mass spectrometry (ESI-MS/MS) in positive ion mode. After chromatography, ESI-MS/MS was carried out using a Q Exactive™ triple quadrupole instrument (Thermo Fisher Scientific, Waltham, MA, USA) equipped with an ESI source. By analysis and comparing secondary fragments of peptides from the collision-induced dissociation spectrum of the protonated molecule [M + H]^+^ in the Uniprot database, the sequences of characteristic peptides were determined. The conditions of mass spectrometer were followed: Resolution: 75,000, AGC target:1e5, maximum IT: 40 ms, Top N: 20, NCE/stepped NCE: 27.

### 4.4. Amino Acid Composition of OPs

The composition and content of the amino acids of OPs were detected by an automatic analyzer. According to our previous method with a slight modification [42], 10 mL of 6 mol/L HCl was added to a hydrolysis tube containing phenol. After vacuuming, the mixture was purged with nitrogen and hydrolyzed at 110 °C for 24 h. After cooling, the filtrate and standard samples of amino acids were loaded to the amino acid analyzer. Then, contents of amino acids in the sample could be determined by reading off the standard curve. As for the determination of tryptophan, the sample was hydrolyzed by 6 mol/L NaOH instead of HCL.

### 4.5. Animals and Grouping

Female Kunming (KM) mice (7–8 weeks old) in this study were purchased from Pengyue Experimental Animal Breeding Co., Ltd. (Jinan, China), production license no. 370092000018921. Animals were kept in a specific pathogen-free environment of temperature (23 ± 2 °C), humidity of 55% ± 10% and a 12 h light/dark cycle, housed in standard cages (four mice per cage) and given free access to standard laboratory diet and water. Prior to the start of the experiment, all animals were acclimatized for at least seven days. In this study, an area of 2.5 × 3 cm dorsal skin surfaces of the mice were shaved with razors (Philips, Cixi, China), which were kept hairless throughout the whole experimental period. The experimental protocol was approved by the Animal Care and Use Committee of Guangdong Ocean University, China (no. IACUC-20190107-02), and conducted strictly in compliance with the National Institute of Health guidelines for the care and use of laboratory animals (Chinese Council).

For the in vivo experiment, 48 mice were randomly divided into six groups (eight mice in each group) as follows: (1) CT ( control group): mice not exposed to UVB and administered with 100 µL vehicle (deionized water); (2) UVB (model group): dorsal skin of mice topically applied with 100 µL vehicle after UVB exposure; (3) UVB + OPs-L: dorsal skin of mice topically applied with low dose of APs (0.2 g/kg∙bw bodyweight) after UVB exposure; (4) UVB + OPs-M: dorsal skin of mice topically applied with middle dose of APs (0.4 g/kg∙bw) after UVB exposure; (5) UVB + OPs-H: dorsal skin of mice topically applied with high dose of APs (0.8 g/kg∙bw) after UVB exposure; (6) UVB + VC (positive control group): dorsal skin of mice topically applied with Vitamin C (0.4 g/kg∙bw) after UVB exposure. In this study, 30 min after each UVB exposure, Vehicle, OPs, VC were topically applied on the shaved dorsal area of until no residue was seen. To ensure the peptides fully absorbed into the skin, after the experiment, mice were given enough space and time to move on their own before being returned to the home cage. Vitamin C was chosen as the positive agent since several studies have reported vitamin C can protect against UV-induced skin photo-damage via suppressing the production of ROS and inflammatory cytokines [44,45,46,47,48,49]. The three doses of OPs were set according to a previous study [50].

### 4.6. Preparation of the Photoaged Mouse Model

UVB has been thought to be responsible for the damaging effects in the skin. Therefore, an array of eight UVB lamps (9 W with an emission spectrum between 285 and 350 nm peaked at 310–315 nm, Philips) were used for the construction of the photoaged skin animal model in this study according to previous studies with slight modification [7,25]. Briefly, an array of eight UVB lamps were positioned 16 cm above the floor of a designed apparatus for irradiation. Then, mice were gently placed in the center of the apparatus with their dorsal skin exposed. Mice were exposed to UVB radiation four times a week starting with 40 mJ/cm^2^ each time during the first week, which was subsequently followed by 50 mJ/cm^2^ each time (the second week), then 60 mJ/cm^2^ each time (the third week), and 70 mJ/cm^2^ each time (the fourth week). Exposure was maintained at 70 mJ/cm^2^ each time for the remaining weeks to deliver a total dose of 2.56 J/cm^2^ over the 10 weeks for every mouse. Ultraviolet (UV) radiation intensity and the time length of irradiation exposure could be detected and calculated by a professional UV light meter with UVB probe (Linshang Technology, Shenzhen, China). The dorsal skin of mice was exposed to UVB light at an intensity of 70 mJ/ cm^2^, which was closed to four minimal erythema doses (MED). During the experiment, once blisters and erosion occurred in the dorsal skin of mice, the irradiation was stopped for 2–3 days, which would continue until the symptoms disappeared.

The dorsal skin of mice was photographed under anesthesia at the end of the study. All dorsal skin samples were divided into three parts (an area of 1.0 × 1.0 cm). One part (0.2 g) was quickly cut and precisely weighed for the moisture content test, referring to the previous literature method [25]. Another part was fixed with paraformaldehyde for at least 24 h, dehydrated in a series of graduated ethanol, and embedded in paraffin for future histopathologic detections, while the third part was frozen promptly with liquid nitrogen and kept in a −80 °C refrigerator for further ELISA and Western blotting test.

### 4.7. Histological Examination

Paraffin-embedded dorsal skin tissues were cut into thin slices on slides with a rotary microtome (Leica, Germany), deparaffinized with xylene, and rehydrated through graded alcohols. The degree of skin structure alteration was assessed microscopically by hematoxylin-eosin (H and E). The content and arrangement change of collagen fiber was analyzed by Masson trichrome staining, and elastic fiber alteration was detected by Victoria Blue staining. Staining kits were offered by Solarbio Biotechnology Co. Ltd. (Beijing, China), and all staining methods were conducted in accordance with their respective staining procedures with no modification. Each stained sample was evaluated and quantified through the image analysis program Image Pro Plus 6.0 (Silver Spring, MD, USA). To evaluate the epidermal hyperproliferation following UVB exposure, the thickness of eight randomlyselected locations in epidermis per slide was measured using an optical microscope with 200× magnification (Olympus, Japan).

### 4.8. Determination of SOD, GSH-Px, MDA and Collagen Content in Skin Tissue

Skin tissues (0.2 g) were homogenized (10,000 rpm, 20 s) with nine volumes of deionized water (1.8 mL), and then centrifuged at 12,000 rpm for 5 min. Supernatants were collected to measure the activities of SOD and GSH-Px, and the content of MDA in the skin according to its respective instructions with no modification (Jian Cheng Biological Engineering Institute, Nanjing, China).

Hyp, a characteristic amino acid of collagen, in the skin samples was measured by an Hyp ELISA kit according to the manufacturer’s instructions (Jian Cheng Biological Engineering Institute, Nanjing, China). Collagen content could be converted from Hyp content by multiplying with a conversion factor of 7.46 [51].

### 4.9. Measurement of Pro-Inflammatory Cytokines in Skin Tissue by ELISA Kits

Referring to the kit instructions offered by Shanghai Enzyme-linked Biotechnology Co., Ltd. (Shanghai, China), the concentrations of IL-1β, IL-6, and TNF-α in skin tissues were tested by enzyme-linked immunosorbent assay (ELISA) kits.

### 4.10. Immunohistochemistry Staining

MMP-1and COX-2 expressions in skin tissue were evaluated by immunohistochemistry staining. After skin tissue sections were dewaxed and hydrated, the antigen of skin tissue was recovered by citrate buffer (10 mM, pH 6.0) in a water bath for 3 min, and endogenous peroxidase in tissues was inactivated with 0.3% (*v*/*v*) hydrogen peroxide for 20 min. Subsequently, slides were incubated with mouse polyclonal anti-MPP-1 or anti-COX-2 antibodies (Santa Cruz Biotechnology, Santa Cruz, CA, USA) (1:200) at 4 °C overnight in a wet box. After carefully washing with PBS, sections were incubated with rabbit anti-mouse (1:1000) secondary antibody at room temperature for 30 min, and then washed three times with PBS. Afterwards, 2–3 drops of 3,30-diaminobenzidine (DAB) were added to develop staining colors. After washing, the sections were counterstained with hematoxylin and evaluated under an optical microscope (Olympus Optical Co. Ltd., Tokyo, Japan). Positive cumulative light density of each photo was analyzed by Image-pro plus 6.0 (Media Cybernetics, Inc., Rockville, MD, USA) software.

### 4.11. Western Blotting

According to the manufacturer’s instructions, the total protein of skin tissues was extracted with a commercial kit (Beyotime technology, Shanghai, China). Then concentration of the protein was measured with a BCA kit (Vazyme technology, Nanjing, China). Then, 30 μg of protein samples were loaded onto a 10% polyacrylamide gel, and followed by electrophoresis, transfer to PVDF membranes, and blocking with 5% bovine serum. Subsequently, primary antibodies (diluted 1:500) were added to the PVDF membranes and incubated at 4 °C overnight. After washing, PVDF membranes were incubated with secondary (diluted 1:2000) antibodies at room temperature for 2 h, followed by detection using enhanced chemiluminescence (Wenyuange biotechnology, Shanghai, China). Eventually, the bands were scanned and analyzed with a chemiluminescence system (Tanon 5200, Shanghai, China). The mouse monoclonal primary antibodies and rabbit anti-mouse secondary antibodies were purchased from Santa Cruz Biotechnology, USA.

### 4.12. Statistical Analysis

Results were expressed as the mean ± standard deviation (SD) and analyzed by SPSS version 17. Experimental values were evaluated by one-way analysis of variation after following the LSD method as multiple comparisons between groups. Significance was indicated at a *p*-value < 0.05 and a *p*-value < 0.01.

## Figures and Tables

**Figure 1 marinedrugs-18-00288-f001:**
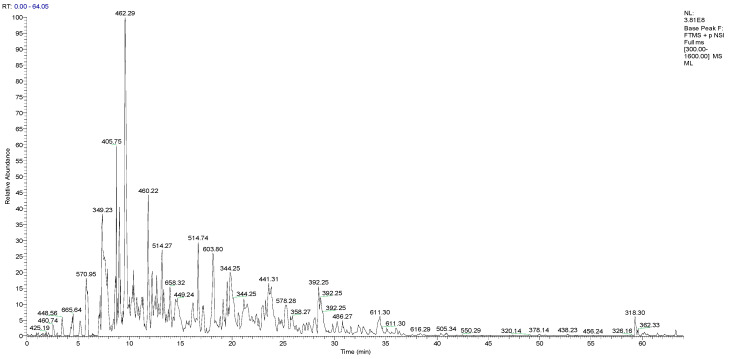
Total ion chromatogram of peptides from oyster enzymatic hydrolysates (OPs).

**Figure 2 marinedrugs-18-00288-f002:**
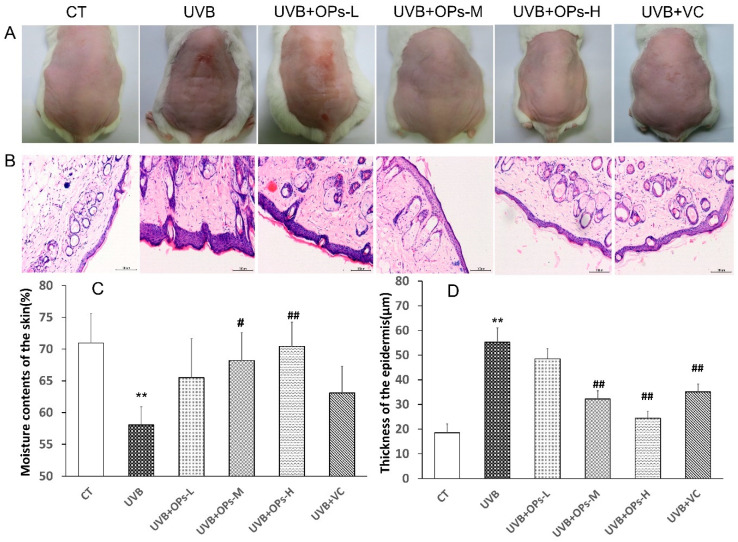
Effects of OPs on UVB-induced skin photodamage in female Kunming mice. (**A**) Visual appearance of mice with different treatments at the end of experiment period. (**B**) H and E staining of dorsal skin sections (200×). Scale bar = 100 μm. (**C**) The moisture content of dorsal skin (n = 8). (**D**) The epidermis thickness of dorsal skin (n = 8). Data represents as means ± standard deviation (SD), ** *p* < 0.01, compared with the CT group; ^#^
*p* < 0.05, ^##^
*p* < 0.01, compared with the UVB group. Scale bar = 100 μm.

**Figure 3 marinedrugs-18-00288-f003:**
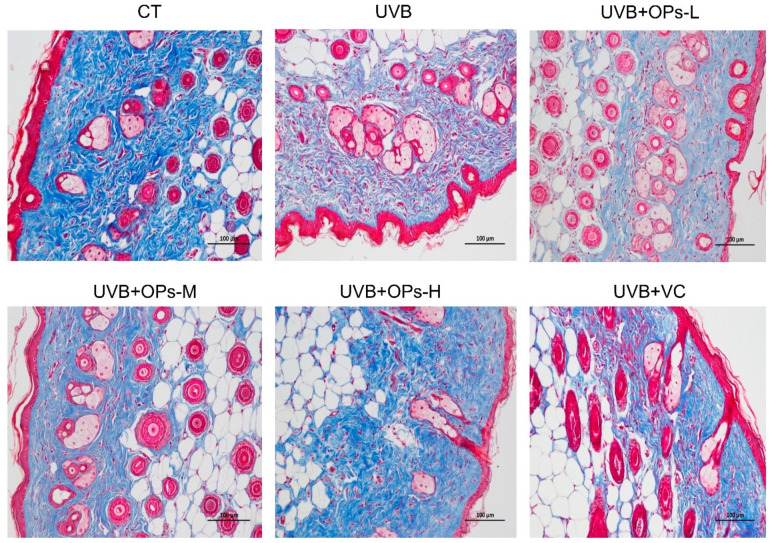
Representative images of Masson’s trichrome-stained dorsal skin section (200×). Slices stained with dark blue indicates collagen fiber, red indicates cytoplasm and muscle fiber. Scale bar = 100 μm.

**Figure 4 marinedrugs-18-00288-f004:**
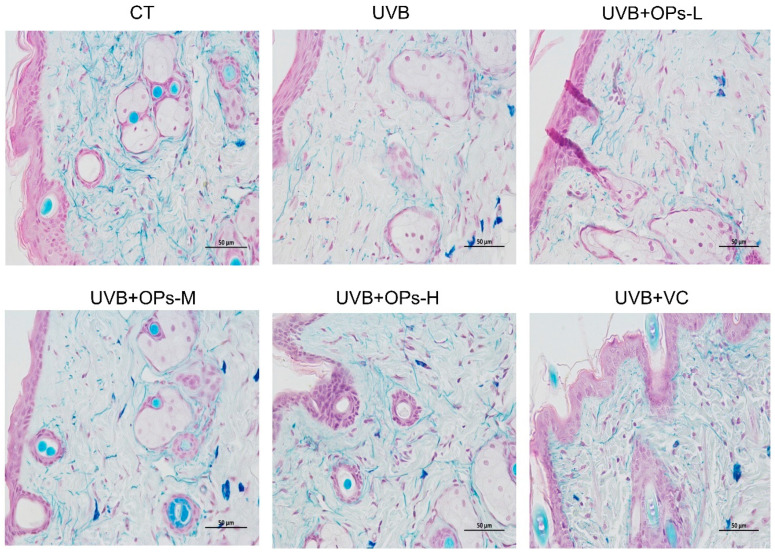
Representative images of Victoria Blue-stained dorsal skin section (400×). Slices stained with light blue indicate collagen fiber. Scale bar = 50 μm.

**Figure 5 marinedrugs-18-00288-f005:**
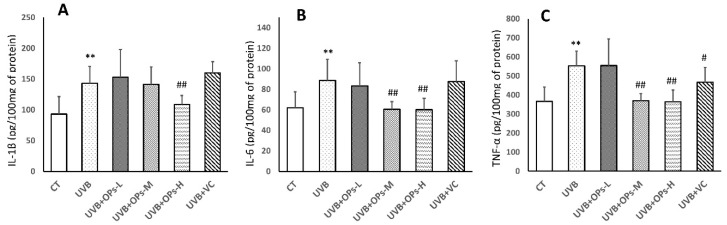
The effect of OPs on pro-inflammatory cytokines contents. (**A**) IL-1β concentration in the dorsal skin. (**B**) IL-6 concentration in the dorsal skin. (**C**) TNF-α concentration in the dorsal skin. data was expressed as mean ± SD (n = 8). ** *p* < 0.01, compared with the CT group; ^#^
*p* < 0.05 and ^##^
*p* < 0.01 compared with the UVB group.

**Figure 6 marinedrugs-18-00288-f006:**
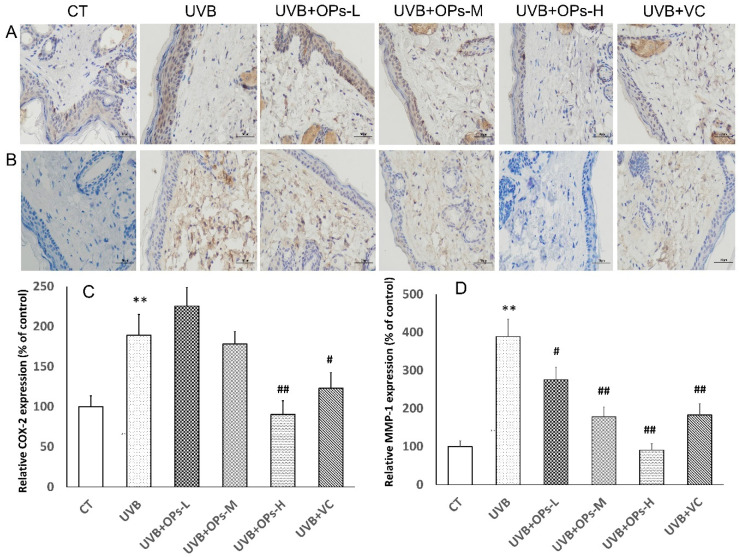
The effects of OPs on UVB-induced epidermal cyclooxygenase-2 (COX-2) and dermic matrix metalloproteinase-1 (MMP-1) expressions in dorsal skin of mice. (**A**) Representative photographs of epidermal COX-2 distribution by IHC staining (400×); Scale bar = 50 μm. (**B**) Representative photographs of epidermal MMP-1 distribution by IHC staining (400×); Scale bar = 50 μm. (**C**) Relative COX-2 positive expression in epidermal layer was assessed by comparing the number of COX-2 positive cells in different groups vs. in the control group from 10 equal sections of immune-stained dorsal skin per animal (n = 4). (**D**) Relative MMP-1 positive expression in dermis layer was assessed by comparing the number of COX-2 positive cells in different groups vs. in the control group from 10 equal sections of immune-stained dorsal skin per animal (n = 4). ** *p* < 0.01, compared with the CT group; ^#^
*p* < 0.05 and ^##^
*p* < 0.01 compared with the UVB group.

**Figure 7 marinedrugs-18-00288-f007:**
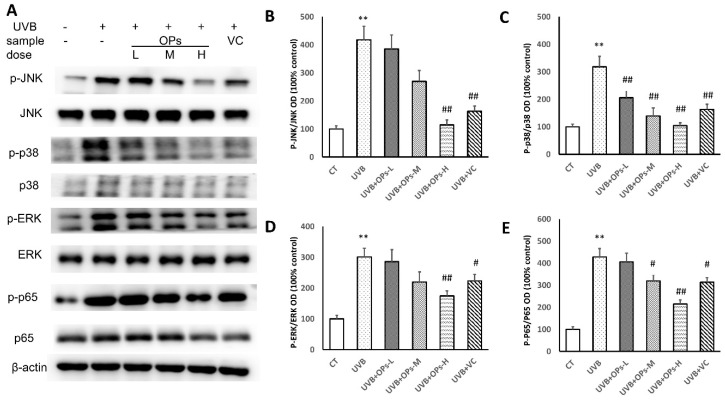
Protein expression levels of MAPKs and NF-κB (p65) in the skin. (**A**) Representative western blot images of MAPK (p-JNK, p-p38, p-ERK,) and NF-κB (p65) signal intensities from multiple experiments. (**B**) Quantitative densitometric analysis of p-JNK/JNK bands. (**C**) Quantitative densitometric analysis of p-p38/p38 bands. (**D**) Quantitative densitometric analysis of p-ERK/ERK bands. (**E)** Quantitative densitometric analysis of p-p65/p65 bands. All data are presented as the mean ± SEM of four independent experiments, ** *p* < 0.01, compared with the CT group; ^#^
*p* < 0.05 and ^##^
*p* < 0.01 compared with the UVB group.

**Figure 8 marinedrugs-18-00288-f008:**
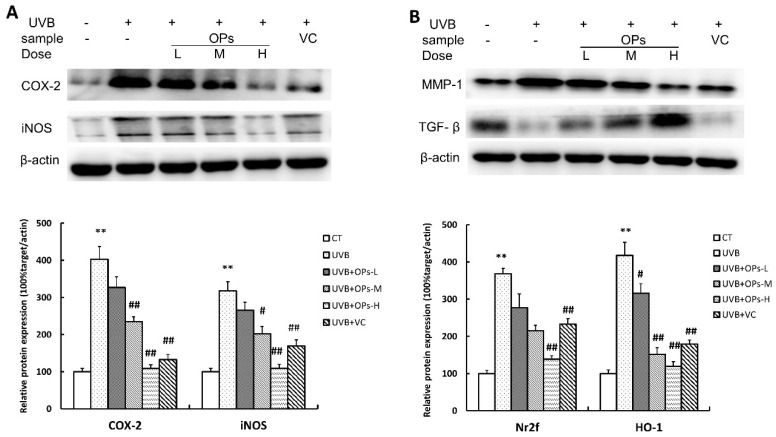
Protein expressions of COX-2, iNOS, MMP-1, and TGF-β in the dorsal skin of mice by Western blotting. (**A**) Protein expressions of COX-2, and iNOS in dorsal skins of mice; (**B**) Protein expressions of MMP-1 and TGF-β in dorsal skins of mice. Data were normalized to protein expression of housekeeping gene beta-actin using the delta-Ct method, and expressed as mean ± S.E.M fold of control. ** *p* < 0.01, compared with the CT group; ^#^
*p* < 0.05 and ^##^
*p* < 0.01 compared with the UVB group.

**Figure 9 marinedrugs-18-00288-f009:**
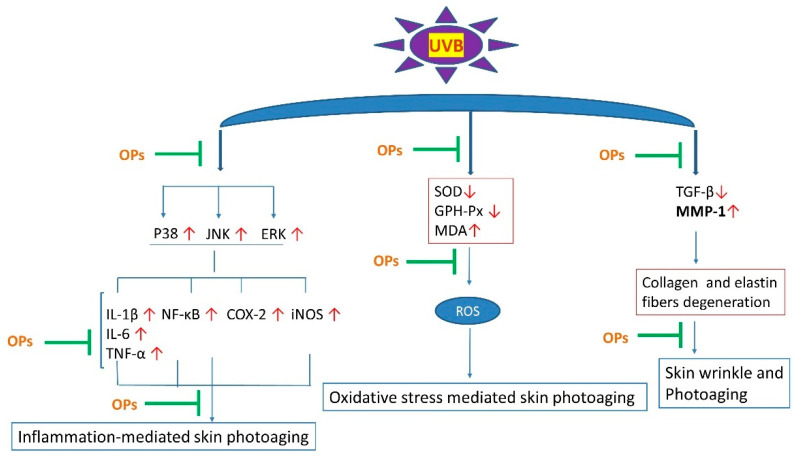
Possible mechanism by which OPs attenuated UVB-induced skin photoaging. ↑: level was increased, ↓level was decreased. green color represents effective.

**Table 1 marinedrugs-18-00288-t001:** Main peptide sequences of OPs.

Sequence	Peptide Sequence	Length	Molecular Mass (Da)	m/z	Scores
1	LTDDQVDEIIRN	12	1429.70	714.85	257.09
2	LTDDQVDEIIRNT	13	1530.75	765.37	144.7
3	LTDDQVDEIIR	11	1315.66	657.83	132.32
4	MWEGEEPTPSEGGPTPK	17	1827.79	913.89	260.22
5	WEGEEPTPSEGGPTPK	16	1696.75	848.37	215.77
6	NNDDIEGSPFK	11	1234.54	617.27	169.58
7	SIDVVILDPH	10	1106.59	553.29	167.9
8	MHIPGSPFE	9	1013.46	506.73	162.31
9	LCPDWEDWNPKN	12	1572.66	786.33	159.79
10	INCKDNRDGTCT	12	1452.60	726.30	146.11
11	IVENPDGTFS	10	1077.49	538.74	137.13
12	VEYLPSKPGEYD	12	1395.65	697.82	133.91
13	YNNDDIEGSPFK	12	1397.60	698.80	131.17
14	LTDGNGRDVPVKT	13	1370.71	685.35	126.71
15	FGKDPFGKDPFDKD	14	1611.75	805.87	210.93
16	GKDPFGKDPFDKD	13	1464.68	732.34	174.57
17	LFQL	4	519.30	519.30	94.737
18	FDLEL	5	635.31	635.31	86.627
19	LLLE	4	486.30	486.30	83.129
20	LLDP	4	456.25	456.25	82.602
21	LEKNKDPINEN	11	1312.66	656.33	233.84
22	ILEEECMFPK	10	1294.59	647.29	144.09
23	FSCRCNCDGSWNCPS	15	1905.66	952.83	205.31
24	SCRCNCDGSWNCPSS	15	1845.62	922.81	179.5

Note: Scores are obtained by comparing a known protein database to measure the similarity between theoretical and experimental mass spectra.

**Table 2 marinedrugs-18-00288-t002:** Composition and contents of amino acids of OPs.

Amino Acid	Total Amino Acid (g/100 g)	Free Amino Acid (g/100 g)	Hydrolyzed Amino Acid (g/100 g)
Aspartic Acid	2.18	Methionine *	0.23
Threonine *	0.94	Isoleucine *	0.94
Serine	0.78	Leucine *	1.56
Glutamic acid	3.59	Tyrosine	1.09
Proline	0.86	Phenylalanine *	0.55
Glycine	1.79	Lysine	1.33
Alanine	1.79	Histidine	0.31
Cysteine	1.17	Arginine *	1.87
Valine *	0.23	Tryptophan *	0.45
Total	21.66		

Note: * essential amino acid.

**Table 3 marinedrugs-18-00288-t003:** Effects of OPs on SOD, and GSH-Px activities, as well as MDA and collagen contents on UVB-induced photoaging mouse skin.

Groups	SOD U/mg protein	GSH-PX U/mg protein	MDA Nmol/mg protein	Collagen (ug/mg protein)
CT	66.02 ± 13.81	249.86 ± 52.79	6.74 ± 0.67	20.11 ± 3.17
UVB	33.05 ± 8.63 *	99.01 ± 37.65 **	25.39 ± 1.23 **	15.58 ± 2.23 *
UVB + OPs-L	48.44 ± 8.89	115.48 ± 35.01	14.95 ± 3.63 ^#^	15.28 ± 3.86
UVB + OPs-M	49.60 ± 8.87 ^#^	140.79 ± 22.50 ^##^	9.77 ± 1.22 ^##^	18.04 ± 3.58 ^#^
UVB + OPs-H	62.86 ± 16.19 ^##^	242.31 ± 39.70 ^##^	9.65 ± 1.32 ^##^	17.67 ± 0.88 ^#^
UVB + VC	52.70 ± 9.63 ^##^	158.48 ± 53.83 ^#^	16.49 + 3.76 ^#^	15.91 ± 2.85

Note: Each value represents the mean ± SD (n = 8). **p* < 0.05, ***p* < 0.01, compared with the CT group; ^#^
*p* < 0.05 and ^##^
*p* < 0.01 compared with the UVB group. One unit of SOD activity was defined as the amount of the enzyme inhibiting the oxidation by 50%. One unit of glutathione peroxidase was defined as the amount of the enzyme leading 1 μmol GSH oxidized per min. SOD, superoxide dismutase; GSH-Px, glutathione peroxidase; MDA, malondialdehyde.

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
