# Peer review of "Ameliorative Effects of Peptides from the Oyster (Crassostrea hongkongensis) Protein Hydrolysates against UVB-Induced Skin Photodamage in Mice"

_marinedrugs, 2020, doi:10.3390/md18060288_

Round 1
Reviewer 1 Report
Peng et al., described an interesting work about the activity of peptides against UV damage extracted by Oyster.
I have some revisione to request:
1) I recommend the authors to study the plasma degradation peptides in vitro in order to understand their plasma half- life. I suggested to study this by HPLC
2) The peptides are uses at a concentration of 50 mg/mL that is very high. Do you try lower concentrations? In this context could be important to perform some Cytotoxic assays to extablish the no Cytotoxic concentration that can be also active.
3) for the lenght these molecules are peptide and not small molecules, I suggest to call them only peptides.
4) Manuscript need an english restyle
Author Response
Response to Reviewer 1 Comments
Thank you for giving us the opportunity to revise our manuscript. For your convenience, all revised items are now highlighted in red colour in the resubmitted manuscript text file.
Point 1: Peng et al., described an interesting work about the activity of peptides against UV damage extracted by Oyster. 

Response 1: Thanks very much for your positive comments.
Point 2: I have some revision to request:
1) I recommend the authors to study the plasma degradation peptides in vitro in order to understand their plasma half- life. I suggested to study this by HPLC
Response 2: Thanks very much for the reviewer’s kind advice. Indeed, plasma half-life should be measured when we evaluate the ability and effect of food-derived bioactive peptides through oral administration. According to the literatures, the method of measuring the plasma half- life by HPLC in vitro was generally applied for peptides with determined sequences. Considering that the sample we used in this study is a mixture of peptides from oyster protein hydrolytes, and the sequences of characteristic peptides was just determined by comparing the secondary mass spectrometer from ESI-MS/MS in the Uniprot database, which requires further confirmation, as well as the delivery method of this sample was through topical application on the skin, thus, we didn’t measure the plasma half- life in this study. However, in future study, we may apply this efficient HPLC approach for the optimization of the plasma stability of therapeutic peptides as suggested.
Point 3: 2) The peptides are uses at a concentration of 50 mg/mL that is very high. Do you try lower concentrations? In this context could be important to perform some Cytotoxic assays to extablish the no Cytotoxic concentration that can be also active. 

Response 3: In our previous manuscript, the doses of peptides sample we used this study were not clearly described. Thus, this part was rewritten in the resubmitted manuscript (Lines427-434). In this study, oyster peptides (OPs) sample with a concentration of 50 mg/mL (about 100 µL was applied for each mouse) was used in the low dose treatment group. The dose of OPs in the UVB+OPs -L group should be 0.2 g/kg.bw but not the concentration of 50 mg/mL. Since topical application of OPs at the dose of 0.2 g/kg showed limited effect on UVB-induced photodamaged skin, we did not try lower doses. Before in vivo study, we performed cytotoxicity test of OPs on human keratinocyte line (Hacat cells), and OPs showed no cytotoxicity with concentrations ranged from 10 µg/ml to 1600 µg/ml. To make readers focus more on the anti-photoaging effects of OPs in vivo, we did not put this result into this paper.
Point 4: 3) for the length these molecules are peptide and not small molecules, I suggest to call them only peptides. 

Response 4: It has been corrected to “peptides” in our resubmitted manuscript.
Point 5: 4) Manuscript need an English restyle.
Response 5: According to the reviewer’s suggestion, we carefully bushed up the language of manuscript and made some changes in the resubmitted manuscript. These changes will not influence the content and framework of this work. Here, we did not list the changes but marked in red in revised manuscript.
Reviewer 2 Report
The research described in this manuscript is interesting, both for topic and for experimental design.
The introduction is well structured, the description of results is good as well tables and figures, the references are sufficiently updated. A
Abstract describes the whole study well.
Therefore, as far as I am concerned, the manuscript can be published on Marine Drugs, except for minor revisions, listed below:
Lines 118-128. This part describing the treatment of mice should be better inserted in the materials and methods section, paragraph 4.5, because it is not a result.
Line 205-206: The mode of collagen content calculating is described in the materials and methods section. These two lines in the context of the results are useless.
Lines 477-478: 477. 4-5 drops of ready-to-use goat serum goat serum. this sentence seems to be incomplete
Author Response
Response to Reviewer 2 Comments
Thank you for giving us the opportunity to revise our manuscript. For your convenience, all revised items are now highlighted in red colour in the resubmitted manuscript text file.
Point 1: The research described in this manuscript is interesting, both for topic and for experimental design. The introduction is well structured, the description of results is good as well tables and figures, the references are sufficiently updated. Abstract describes the whole study well. Therefore, as far as I am concerned, the manuscript can be published on Marine Drugs, except for minor revisions, listed below: 

Response 1: Thanks very much for your positive comments.
Point 2: Lines 118-128. This part describing the treatment of mice should be better inserted in the materials and methods section, paragraph 4.5, because it is not a result.
Response 2: It has been modified in the resubmitted manuscript.
Point 3: Line 205-206: The mode of collagen content calculating is described in the materials and methods section. These two lines in the context of the results are useless.

Response 3: It has been deleted.
Point 4: Lines 477-478: 477. 4-5 drops of ready-to-use goat serum goat serum. this sentence seems to be incomplete

Response 4: It is our negligence and we are sorry about this. It has been deleted (Lines 492-493).
Reviewer 3 Report
In this article, authors evaluated the photo-protective effect of small peptides from oyster protein hydrolysates by topical application on the skin of mice exposed to UVB irradiation. First, the small peptides were purified and characterized from a chemical point of view. Then, the beneficial effects of the topical application of these peptides were studied upon irradiation of mice with UVB. The topical application of the peptides on the UVB exposed skin significantly alleviated dehydration, epidermal hyperplasia and degradation of collagen and elastin fibers. Moreover, the highest doses of the treatment revealed the increase of the antioxidant enzymes (SOD and GPH-Px) activities, while decreased the malondialdehyde (MDA) level in the skin. Several inflammatory markers resulted also decreased: cytokines (IL-1β, IL-6, TNF-α), iNOS and COX-2. Finally, the authors suggested that the possible molecular mechanism underlying the anti-photo-aging of the peptides may be related to downregulation of MAPK/ NF-κB signaling pathway, while promoting TGF-β production in the skin.
Overall, the manuscript is mostly well-written, clear and informative. The results section contains a comprehensive study from the histological observations of the efficacy of the treatments applied down to the molecular markers of inflammation, oxidative stress and photo-aging.
The bibliography reported in the article is also ample and supports the evidences described in the results and discussion sections.
The article reaches the quality standards of this Journal for publication with only minor revisions according to the comments reported here below:
- It would be interesting in the article to add comparisons of the UVB dose administered to mice with the natural expositions to UVB light. For instance, how many hours of sunbath or tanning booths does the dose of UVB administered to mice correspond to?
- In Table 1, the scores were obtained by scoring a known protein database to measure the similarity between theoretical mass spectra and experimental mass spectra. But what about the sequences? Are the sequence also verified or the reported sequences are only an example? The presented peptides may bear the same aminoacids but with a different sequence? Please explain.
- For the in vivo experiments, one important control group is missing: the group of mice exposed to SMP but not receiving the UV B. This group should be interesting to understand if the results described are correlated to the exposure of the skin to the treatment itself or correlated to the treatment UPON UVB exposure. Toxicity effects may also be depicted by this group.
- I suggest to join all the results concerning COX-2 and MMP-1 together in one figure: the IHS of Figure 6 and the WB of figure 8.
- In the material and method section, please add a paragraph describing how the topical treatment was applied and how the mice were kept away from each other to prevent the swept out of the treatment. Please also explain how and when the treatments were administered. Are the treatment applied before and/or after UVB exposure? Are the treatments applied every time before and/or after the UVB exposure? Are the doses reported equal to one administration or is it the total dose received?
- Line 75: you can modify “People” with “people”
- Line 97: are you sure you want to indicate the value 0 for the molecular peptides?
- Line 316: is ICH or IHC?
Author Response
Response to Reviewer 3 Comments
Thank you for giving us the opportunity to revise our manuscript. For your convenience, all revised items are now highlighted in red colour in the resubmitted manuscript text file.
Point 1: In this article, authors evaluated the photo-protective effect of small peptides from oyster protein hydrolysates by topical application on the skin of mice exposed to UVB irradiation. First, the small peptides were purified and characterized from a chemical point of view. Then, the beneficial effects of the topical application of these peptides were studied upon irradiation of mice with UVB. The topical application of the peptides on the UVB exposed skin significantly alleviated dehydration, epidermal hyperplasia and degradation of collagen and elastin fibers. Moreover, the highest doses of the treatment revealed the increase of the antioxidant enzymes (SOD and GPH-Px) activities, while decreased the malondialdehyde (MDA) level in the skin. Several inflammatory markers resulted also decreased: cytokines (IL-1β, IL-6, TNF-α), iNOS and COX-2. Finally, the authors suggested that the possible molecular mechanism underlying the anti-photo-aging of the peptides may be related to downregulation of MAPK/ NF-κB signaling pathway, while promoting TGF-β production in the skin.
Overall, the manuscript is mostly well-written, clear and informative. The results section contains a comprehensive study from the histological observations of the efficacy of the treatments applied down to the molecular markers of inflammation, oxidative stress and photo-aging.
The bibliography reported in the article is also ample and supports the evidences described in the results and discussion sections.
The article reaches the quality standards of this Journal for publication with only minor revisions according to the comments reported here below:

Response 1: Thanks very much for your positive comments.
Point 2: It would be interesting in the article to add comparisons of the UVB dose administered to mice with the natural expositions to UVB light. For instance, how many hours of sunbath or tanning booths does the dose of UVB administered to mice correspond to?
Response 2: Thanks very much for your advice. In this study, in order to prepare a photodamaged mice model, an array of 8 UVB lamps were positioned 16 cm above the floor of a designed apparatus for irradiation. Then, mice were gently placed in the central of apparatus with dorsal skin exposed. Ultraviolet (UV) radiation intensity and the time length of each irradiation exposure could be detected and calculated by a professional UV light meter with UVB probe. Minimal erythema dose (MED) of UVB on Kunming could also be tested by this UV light meter. In this study, the dorsal skin of mouse exposed to UVB irradiation at an intensity of 70 mJ/ cm2 was closed to four MEDs, which lasted 10 minutes. To help reader understand this information clearly, we added more details about the UVB irradiation procedure in our revised manuscript (Lines 444-454).
Since the energy power unit (W/cm2) of sunlight and tanning machine varies in different place and season, as well as with different equipment, we did not compare the UVB dose administered to mice with the natural expositions to UVB light.
Point 3: In Table 1, the scores were obtained by scoring a known protein database to measure the similarity between theoretical mass spectra and experimental mass spectra. But what about the sequences? Are the sequence also verified or the reported sequences are only an example? The presented peptides may bear the same amino acids but with a different sequence? Please explain.
Response 3: Peptides sequences were identified by electrospray ionization mass spectrometry and tandem mass spectrometry (ESI-MS/MS) in positive ion mode. After chromatography, ESI-MS/MS was carried out using a triple quadrupole instrument Q Exactive™ (Thermo Scientific) equipped with an ESI source. By analysis and comparing secondary fragments of peptides from the collision-induced dissociation spectrum of the protonated molecule [M +H] + in the Uniprot database, the sequences of characteristic peptides were determined. Those details were added into materials and methods of the revised manuscript (Lines 398-405), and our previous publication with this method has been also cited.
As discussed above, sequences of characteristic peptide were just determined by comparing the secondary mass spectrometer from ESI-MS/MS in the Uniprot database, which requires further confirmation by subsequent separation and purification and identification.
In this study, we not only compared the molecular weight of the complete peptide the from primary mass spectrometer total ion map, but also performed secondary fragmentation to break the polypeptide into smaller peptides. Moreover, the scores in this study on one hand reflect the similarity between theoretical mass spectra and experimental mass spectra, on the other hand, higher scores represent lower probability of peptides may bear the same amino acids but with a different sequence. According to analysis of comparing software, provided its score was higher than 60, sequence of peptide would be considered reliable. we have added this information in our revised manuscript (Lines 107-108).
Point 4: For the in vivo experiments, one important control group is missing: the group of mice exposed to SMP but not receiving the UV B. This group should be interesting to understand if the results described are correlated to the exposure of the skin to the treatment itself or correlated to the treatment UPON UVB exposure. Toxicity effects may also be depicted by this group. 

Response 4: The reviewer provided us a good suggestion. Considering that peptides we used in this study are food-derived peptides, and its extraction method is relatively moderate, we did not set the group of mice exposed to peptides for oyster but not receiving the UVB irradiation. In future, if we further investigate the anti-photoaging effect of purified active peptide from this oyster peptide sample. we will set this control group and perform toxicity experiment, as reviewer suggested.
Point 5: In the material and method section, please add a paragraph describing how the topical treatment was applied and how the mice were kept away from each other to prevent the swept out of the treatment. Please also explain how and when the treatments were administered. Are the treatment applied before and/or after UVB exposure? Are the treatments applied every time before and/or after the UVB exposure? Are the doses reported equal to one administration or is it the total dose received?
Response 5: According to reviewer’s advice, we have added more details of treatment and UVB irradiation in the material and method section (Lines 427-456).
Point 6: I suggest to join all the results concerning COX-2 and MMP-1 together in one figure: the IHS of Figure 6 and the WB of figure 8.
Response 6: As the reviewer suggested, joining all the results concerning COX-2 and MMP-1 together in one figure may clearly show the changes of COX-2 and MMP-1 at histo- pathology, and the protein expression level. However, in this paper, we put the results from IHC and WB in different figures due to following considerations. Firstly, COX-2 and iNOS are two important pro-inflammatory enzymes, as well as two important markers for inflammation. By putting the WB results of COX-2 and iNOS together in figure 8A, the effect of oyster peptides on the expression pro-inflammatory enzymes can been clearly seen. Secondly, MMP-1 overexpression was reported to downregulate the activity of transforming growth factor beta (TGF-β), and TGF-β/Smad pathway is a major pathway responsible for procollagen production. Thus, putting the WB results of MMP-1 and TGF-β together in figure 8B. he effect of oyster peptides regulating MMP-1 the expression and TGF-β pathway can been clearly seen.
Point 7: Line 75: you can modify “People” with “people”
Response 7: It has been modified (Line 75)
Point 8: Line 97: are you sure you want to indicate the value 0 for the molecular peptides?
Response 8: It has been corrected as “below 2000 Da” in our resubmitted manuscript (Line 97).
Point 9: Line 316: is ICH or IHC?
Response 9: It is IHC, the abbreviation of immunohistochemical
Round 2
Reviewer 1 Report
The authors replied in an exhaustive mode and they added some important infomations. I suggest them to study better the peptide sequences for the next works with a deepest structural characterization.
Author Response
Response to Reviewer 1 Comments
Thank you for giving us the opportunity to revise our manuscript again. For your convenience, all revised items are now highlighted in yellow colour in the resubmitted manuscript text file.
Point 1: The authors replied in an exhaustive mode and they added some important infomations. I suggest them to study better the peptide sequences for the next works with a deepest structural characterization.

Response 1: Thank you very much for your comments on our manuscript. In the present study, we mainly evaluated the ability and efficiency of topical application of peptides from the oyster protein hydrolysates (OPs) in an UVB-induced photodamaged mice model. Next step, as you suggested, we will purify these peptides with Sephadex G-25 column and RP-C18 column, further determine sequences with UPLC-MS/MS. Hopefully, a deepest structural characterization of these peptide sequences would be confirmed, and these results will be further reported in our following papers.